# Autophagy-Like Cell Death Regulates Hydrogen Peroxide and Calcium Ion Distribution in *Xa3/Xa26*-Mediated Resistance to *Xanthomonas oryzae* pv. *oryzae*

**DOI:** 10.3390/ijms21010194

**Published:** 2019-12-27

**Authors:** Jianbo Cao, Meng Zhang, Mengmeng Zhu, Limin He, Jinghua Xiao, Xianghua Li, Meng Yuan

**Affiliations:** 1National Key Laboratory of Crop Genetic Improvement, National Center of Plant Gene Research (Wuhan), Huazhong Agricultural University, Wuhan 430070, China; mengzhang@webmail.hzau.edu.cn (M.Z.); zhumengxv@yeah.net (M.Z.); xiaojh@mail.hzau.edu.cn (J.X.); xhli@mail.hzau.edu.cn (X.L.); 2Public Laboratory of Electron Microscopy, Huazhong Agricultural University, Wuhan 430070, China; limin_room@163.com

**Keywords:** *Xa3*/*Xa26*, autophagy-like cell death, hydrogen peroxide, calcium ion, ultrastructure

## Abstract

The broad-spectrum and durable resistance gene *Xa3/Xa26* against *Xanthomonas oryzae* pv. *oryzae* (*Xoo*) has been widely exploited in rice production in China. But the cytological features of the *Xa3/Xa26-*mediated resistance reaction have been rarely reported. This study reveals the cytological characteristics of the *Xa3/Xa26*-mediated resistance reaction against *Xoo* to uncover the functions of hypersensitive response programmed cell death (HR-PCD) in rice. Autophagy-like cell death, which was characterized by double-membrane bodies appearance in xylem parenchyma cell and mesophyll cell, was inhibited by autophagy inhibitor 3-methyladenin (3-MA). The autophagy-related genes were induced to reach a high level in resistance reaction. The hydrogen peroxide (H_2_O_2_) maintained a low concentration on the plasma membrane. The calcium ions localized on the apoplast were transferred into the vacuole. The autophagy inhibitor (3-MA) impaired *Xa3/Xa26-*mediated resistance by promoting the accumulation of H_2_O_2_, and inhibited the transfer of extracellular calcium ions into the vacuole in the xylem parenchyma cells and mesophyll cells. Therefore, the HR-PCD belongs to autophagy-like cell death in the *Xa3/Xa26-*mediated resistance reaction. These results suggest that the autophagy-like cell death participates in the *Xa3/Xa26*-mediated resistance by negatively regulating H_2_O_2_ accumulation, in order to abolish oxidative stress and possibly activate calcium ion signals in xylem parenchyma cells of the rice leaf.

## 1. Introduction

The bacterial blight caused by *Xanthomonas oryzae* pv. *oryzae* (*Xoo*) is the most serious bacterial disease in rice (*Oryza sativa*) worldwide. Major disease resistance (*MR*) genes, which are of qualitative or complete resistance against *Xoo*, are currently used in rice breeding programs [1,2]. Approximately 41 *MR* genes against *Xoo* have been identified, and 11 (*Xa1*, *Xa3/Xa26*, *Xa4*, *xa5*, *Xa10*, *xa13*, *Xa21*, *Xa23*, *xa25*, *Xa27*, and *xa41*) of these 41 *MR* genes have been characterized at present [1]. However, the rapid loss of resistance in rice varieties carrying a single *MR* gene is a big problem for breeders and cultivators [3]. Therefore, it is necessary to comprehend *MR* gene-mediated resistance mechanisms in rice [4].

Like *Arabidopsis*, rice has a two-tiered innate immune system: pathogen-associated molecular pattern-triggered immunity (PTI) (also called plant-derived damage-associated molecular pattern-triggered immunity, or basal resistance), and effector-triggered immunity (ETI) (also referred to gene-for-gene resistance) [1,2]. PTI, the first layer of defense, is initiated by plasma membrane-localized rice pattern recognition receptors (PRRs), which are receptor-kinase proteins or receptor-like proteins [2]. ETI, the second layer of rapid and robust defense, is triggered by intracellular *MR* proteins (immune receptors) recognizing the pathogen effector molecules [2]. The defense response of ETI is usually associated with the local hypersensitive response-programmed cell death (HR-PCD) which is characterized by the local and rapid cell death at sites of pathogen attack [3]. However, *MR* gene-mediated resistance against *Xoo* cannot be singly explained by the mechanism of PTI or ETI [1].

HR-PCD is often, but not always, a part of ETI initiated by cytoplasmic nucleotide-binding (NB)-leucine rich repeat (LRR)-type proteins in *Arabidopsis* [5]. HR-PCD of *Arabidopsis* always exhibits the autophagy characterized by autophagosomes in cytoplasm and the vacuole-mediated cell death characterized by tonoplast disruption at morphological levels [6,7]. The autophagosomes have the following ultrastructure: (1) the tonoplast-bound bodies (microautophagy) originated from cytoplasm, cytoplasmic bodies, or organelles in the vacuole; (2) the double-membrane bodies (macroautophagy) derived from cytoplasm substance bound by an endoplasmic reticulum-like tubule in the cytoplasm; (3) the multilamellar bodies generated from many membranes bound by a single membrane in the cytoplasm [8]. Furthermore, the dominant and recessive *MR* genes lead to different PCDs in the rice resistance reaction to *Xoo* [9]. The dominant *Xa1*-, *Xa4*-, and *Xa21*-mediated resistances are mainly associated with autophagy-like cell death demonstrated by the formation of autophagosome-like bodies in the xylem parenchyma cells of the rice leaf. Autophagosome-like bodies in rice are similar to the autophagosomes in *Arabidopsis* [9]. In contrast, the recessive *xa5*- and *xa13*-mediated resistances are mainly associated with the destructive vacuolar-mediated cell death characterized by tonoplast disruption in xylem parenchyma cells [9]. The autophagy inhibitor 3-methyladenine (3-MA) can partially impair the *Xa1*-, *Xa4*-, and *Xa21*-mediated resistances, just as an alkaline solution does to *xa5*- and *xa13*-mediated resistances [9].

HR-PCD can restrict pathogen proliferation by blocking the biotrophic pathogens from acquiring nutrients of plant cells, and suppress hemi-biotrophic and necrotrophic pathogens by possibly releasing vacuole toxins [6,10]. Meanwhile, HR-PCD can also activate plant system acquired immunity and resistance reaction of cells and tissues neighboring to the infected cells and tissues [11]. Autophagy is reported to participate in the resistance to the tobacco mosaic virus in *Nicotiana benthamiana,* as well as the resistance to the bacterial pathogen *Pseudomonas syringae* pathovar (pv) in tomato (*Pst*) in the *Arabidopsis*, through the pro-survival pathway, restricting HR-PCD [12,13]. However, another previous study has also reported different findings, namely that autophagy only executes HR-PCD in *Arabidopsis* resistance against *Pst* [14]. Therefore, it is controversial whether autophagy restricts cell death via the pro-survival pathway or executes cell death at the infection site via pro-death pathways in the *Arabidopsis* resistance against *Pst* [12,13,14].

The first observed phenomenon in HR-PCD is an oxidative burst and the generation of reactive oxygen species (ROS), including singlet oxygen, superoxide, radical hydroxyl, and hydrogen peroxide (H_2_O_2_) [15]. H_2_O_2_ is the major and most stable type among ROS in plants [15]. ROS mainly originates from nicotinamide adenine dinucleotide phosphate-oxidases on the plasma membrane, and peroxidases on the cell wall, apoplastic polyamine oxidases, chloroplast, peroxisome, and mitochondrion [15,16,17,18]. Meanwhile, ROS are regulated by the reactive oxygen species scavenger system and salicylic acid, nitric oxide, calcium ion, ethylene, and other signal messengers [18]. ROS, as essential signal messengers triggering HR-PCD, can reinforce the cell wall and induce defense-related gene expression [5,15,19,20]. After endoplasmic reticulum (ER) Ca^2+^ depletion, XA10, as the *Xa10* gene product, activates ROS generation in chloroplast in rice resistance reaction against *Xoo* [21]. In *Xa3*/*Xa26*-mediated resistance against *Xoo*, the activation of OsTPI1.1 (a triosephosphate isomerase) decreases the reduction of NADP^+^ to NADPH to maintain the high level of H_2_O_2_ [22]. However, the high concentration of H_2_O_2_ can damage the plasma membrane, resulting in the death of cells [15,19,20].

The *MR* gene *Xa3/Xa26* was first isolated from the rice cultivar Minghui 63 and is named *Xa26* [23]. Minghui 63 is the female parent of the elite hybrid Shanyou 63, serving as the most widely cultivated hybrid in China over the past three decades [24]. The *MR* gene *Xa3* identified from rice cultivar IRBB3 is identical to *Xa26*, and thus is named *Xa3/Xa26* [25]. The *Xa3/Xa26*, which has broad-spectrum and durable resistance to *Xoo*, has been widely exploited in the rice production of China for a long time [26]. *Xa3/Xa26*-mediated resistance to *Xoo* is influenced by the genetic background and the developmental stage [27]. The *Japonica* background of Mudanjiang 8 and Zhonghua 11 cultivars transformed *Xa3/Xa26* with its native promoter to show whole-growth-stage resistance [27]. However, the *Indica* background of Minghui 63 and IRBB3 cultivars carrying *Xa3/Xa26* only show adult stage (from tilling stage to grain filling stage) resistance [27]. The molecular mechanism of *Xa3/Xa26*-mediated resistance modified by the genetic background and developmental stages is determined by the relatively higher expression level of *Xa3/Xa26* throughout the growth stages in the *Japonica* background, while the *Xa3/Xa26* expression level gradually increases from seedling stage to adult stage in the *Indica* background [27]. *Xa3/Xa26* encoding, with a leucine-rich repeat (LRR) receptor kinase, belongs to a multiple gene family, and it is specifically expressed in the xylem vessel cells of rice [28]. The *Xa3/Xa26* gene family members *NRKe* and *9RKe* have been reported to be constitutively expressed to induce the temperature-sensitive cell death of the rice leaf, further forming lesion mimics [29].

In rice, 33 autophagy-related gene (*ATG*) homologues in the genome are classified into 13 *ATG* subfamilies [30]. The expression of most *OsATG* homologues is regulated by hormones, abiotic and biotic stresses, and nutrient limitation treatments [30]. Some *OsATG*s possibly involve the rice–*Xoo* interaction, and *OsATG10b* protects rice cell against oxidative stress [30,31]. In this study, we analyze cell death, localization of H_2_O_2_ and calcium ion in *Xa3/Xa26-*mediated resistance reaction. Our results suggest that *Xa3/Xa26-*mediated autophagy-like cell death eliminates H_2_O_2_ accumulation and changes the distribution of the calcium ion.

## 2. Results

### 2.1. Autophagy-Like Ultrastructure in Xylem Parenchyma and Mesophyll Cells during Xa3/Xa26-Mediated Resistance Reaction

On Day 3 and 5 after inoculation (DAI) with *Xoo* strain PXO61, the resistant IRBB3 plant leaves with *Xa3/Xa26* exhibited autophagosome-like single membrane bodies with electron-dense materials and autophagosome-like double-membrane bodies in xylem parenchyma cells next to xylem vessels containing bacteria (Figure 1A), while the susceptible near-isogenic line IR24 plant leaves without *Xa3/Xa26* showed protoplast shrinkage of the xylem parenchyma cells approximate to xylem vessels containing bacteria (Figure 1B), which is in contrast to respective leaves at 0 DAI (Figure 1A,B). Furthermore, at 5 DAI, the number of cells with autophagosome-like bodies was 4-fold that of the number of cells with protoplast shrinkage in the IRBB3 plant leaf, but the number of cells containing protoplast shrinkage was 5-fold that of the number of cells containing autophagosome-like bodies in the IR24 plant leaf (Figure 1C). The leaf lesion length of IR24 was larger than that of IRBB3 at 14 DAI (Figure 1D). The mesophyll cells of IRBB3 also showed autophagosome-like bodies and those of IR24 presented protoplast shrinkage at 3 and 5 DAI by infiltration inoculation (Appendix A). In IRBB3 and IR24 plants at 5 DAI, the number of observed mesophyll cells with autophagosome-like bodies was similar to that of xylem parenchyma cells infected with *Xoo* (Appendix A). Many autophagosome-like bodies were observed in xylem parenchyma cells of the resistant Rb49 plant and many of the xylem parenchyma cells with protoplast shrinkage were present in the susceptible Mudanjiang 8 plant at 3 and 5 DAI in the *Japonica* rice varieties (Appendix A).

### 2.2. High Expression of Autophagy-Related Genes in Xa3/Xa26-Mediated Resistance Reaction

Autophagy is a highly conserved vesicular mechanism controlled by autophagy related genes (*ATG*) in all eukaryotes [32]. Some rice autophagy-related genes (*OsATG*) possibly involve the rice–*Xoo* interaction [30]. We further analyzed the expression of some representative *OsATG*s in resistant IRBB3 and susceptible IR24 rice leaves infected with the *Xoo* strain PXO61 and found that the relative expression levels of 11 *OsATG*s changed in both *Xa3/Xa26-*mediated resistance (IRBB3) and susceptible reaction (IR24) (Figure 2). On Day 1 after inoculation (DAI), the expressions of 9 *OsATGs*, except for *OsATG12* and *OsATG1a6*, in resistant rice lines were significantly (*p* < 0.01) higher than those in the susceptible rice line. At 2 and 3 DAI, the expression levels of *OsATG8c*, *OsATG7*, *OsATG5,* and *OsATG6a* were significantly (*p* < 0.01) higher in the resistant plant than in the susceptible plant, Meanwhile, *OsATG6b*, *OsATG8b*, *OsATG9b*, *OsATG12,* and *OsATG18e* were induced to a significantly (*p* < 0.01 and *p* < 0.05) higher level in the resistant plant than in the susceptible plant. At 0 DAI, the expression levels of *OsATG7*, *OsATG5*, *OsATG12,* and *OsATG8c* were significantly higher at *p* < 0.01 in the resistant plant, and the expression levels of *OsATG9b*, *OsATG18e*, *OsATG10a* and *OsATG1a6* were higher at *p* < 0.05 than in the susceptible rice plant. The lesion area (%) of the rice leaves for *OsATG* expression analysis in the susceptible rice (IR24) was larger than that of the resistant rice (IRBB3) (Appendix A). *OsATG5*, *OsATG6a*, *OsATG6b*, *OsATG7*, *OsATG8b* and *OsATG12* had similar expression patterns in *Japonica* background Rb49 (resistant rice) and Mudanjiang 8 (susceptible rice) (Appendix A).

### 2.3. Autophagy Inhibitor 3-Methyladenine Partially Impaired Xa3/Xa26-Mediated Resistance by Reducing Autophagosome-Like Body Formation in Mesophyll Cells

ATG proteins encoded by *ATG* genes are the executors of autophagy [32]. The class III phosphatidyinositol 3-kinase (PI3K) combines with ATG6 and other regulatory proteins to form a protein complex that is essential for the nucleation of autophagosome [32,33]. The autophagy inhibitor 3-methyladenine (3-MA) specifically inhibits PI3K activity and has been widely used to block the autophagy in plant cells [33,34,35]. To investigate whether 3-MA inhibits *Xa3/Xa26*-mediated resistance, we observed the representative water-soaked symptom of inoculation sites on resistant Rb49 leaves infiltrated with the *Xoo* strain PXO61 in 3-MA solution and susceptible Mudanjiang 8 leaves infiltrated with PXO61 in H_2_O solution at 3 DAI (Figure 3A). But few water-soaked symptoms of inoculation sites on Rb49 leaves infiltrated with PXO61 in H_2_O solution and no water-soaked symptoms of inoculation sites on Rb49 leaves infiltrated with only 3-MA solution at 3 DAI (Figure 3A). Compared with that on Rb49 plant leaves infiltrated with PXO61 in H_2_O solution, the number of inoculation sites with water-soaked symptoms markedly increased on Rb49 plant leaves infiltrated with PXO61 in 3-MA solution (Table 1). But the number of inoculation sites with water-soaked symptoms was small and even 0 on Rb49 plant leaves infiltrated with PXO61 in H_2_O solution and with only 3-MA solution (Table 1). However, all the inoculation sites exhibited water-soaked symptoms on the Mudanjiang 8 plant leaves infiltrated with PXO61 in H_2_O solution (Table 1). Ultrastructural analysis of the infiltration site area of leaves was shown by red circles in Figure 3A. There were more representative autophagosome-like bodies (black arrow) in cytoplasm of mesophyll cells surrounded by bacteria in Rb49 plant leaves infiltrated with PXO61 in H_2_O solution than those infiltrated with PXO61 in 3-MA solution (Figure 3B-3, 1). Furthermore, little autophagosome-like bodies were observed in the cytoplasm of mesophyll cells in Rb49 plant leaves infiltrated with only 3-MA solution and Mudanjiang 8 plant leaves infiltrated with PXO61 (Figure 3B-2,4). The number of autophagosome-like bodies per mesophyll cell in Rb49 plant leaves infiltrated with PXO61 in 3-MA solution was significantly (*p* < 0.01) larger than that infiltrated with PXO61 in H_2_O solution (Figure 3B-5).

### 2.4. Autophagy Inhibitor 3-Methyladenine Promoted H_2_O_2_ Accumulation in Impaired Xa3/Xa26-Mediated Resistance Reaction

The H_2_O_2_ specifically catalyzes 3, 3′-diaminobenzidine (DAB) to polymerize into brown polymer, which is stable in most solvents [36]. The brown precipitates of DAB polymer exhibited different color intensities on the inoculation sites on the resistant Rb49 and susceptible Mudanjiang 8 plant leaves infiltrated with different solutions (Figure 4). Brown precipitates on inoculation sites of Rb49 leaves infiltrated with *Xoo* strain PXO61 in 3-MA solution exhibited a higher color intensity than those in H_2_O solution, while they showed similar color intensity to those of the Mudanjiang 8 leaves infiltrated with PXO61 in H_2_O solution (Figure 4A). Low color intensity of brown precipitates was observed on the inoculation sites of Rb49 plant leaves infiltrated with only 3-MA solution (Figure 4A). The color intensity of DAB polymer precipitates was calculated as 255 divided by the gray value of each inoculation site, to represent the H_2_O_2_ content on the inoculation sites (Figure 4B). The H_2_O_2_ content of inoculation sites on Rb49 plant leaves infiltrated with PXO61 in 3-MA solution was significantly (*p* < 0.01) higher than that in H_2_O solution (Figure 4B). In the same rice leaves treated by DAB staining, the percentage of inoculation sites with water-soaked symptoms on Rb49 plant leaves infiltrated with PXO61 in 3-MA solution was remarkably higher than that of the H_2_O solution (Appendix A).

### 2.5. Low H_2_O_2_ Accumulation in Xa3/Xa26-Mediated Resistance and the Partially Impaired Resistance by in Vitro Spraying H_2_O_2_ Treatment

On Day 1, 3, 5 after clip leaf inoculation, the H_2_O_2_ content of the resistant Rb49 plant leaf gradually increased, and the H_2_O_2_ content of the susceptible Mudanjiang 8 plant leaf firstly decreased then increased to a high level (Figure 5A). However, the H_2_O_2_ content of Mudanjiang 8 plant leaf at 5 DAI was significantly higher than that at 0 DAI (*p* < 0.01), and that of Rb49 plant leaf at 5 DAI (*p* < 0.05), respectively (Figure 5A). The H_2_O_2_ content in the Mudanjiang 8 plant leaf was significantly (*p* < 0.05) higher than that in the Rb49 plant leaf at 0, 1, 3 DAI (Figure 5A). At 14 DAI, there were no lesion areas of non-inoculated Rb49 and Mudanjiang 8 plant leaves treated by spraying H_2_O_2_ at 14 DAI (Figure 5B); but lesion areas of Rb49 and Mudanjiang 8 plant leaves, which were inoculated by *Xoo* strain PXO61, treated by in vitro spraying H_2_O_2_ were significantly (*p* < 0.01) larger than those of the leaves untreated by H_2_O_2_ spraying (Figure 5B).

### 2.6. Lower H_2_O_2_ Accumulation on Plasma Membrane of Xylem Parenchyma Cells during Xa3/Xa26-Mediated Resistance

CeCl_3_ can specifically react with H_2_O_2_ to generate cerium peroxide precipitates as high electron-density material under transmission electron microscopy (TEM) for in situ displaying of the distribution of H_2_O_2_ [37]. To analyze the H_2_O_2_ distribution in the xylem parenchyma cells of resistant Rb49 and the susceptible Mudanjiang 8 plant leaves infected with *Xoo* strain PXO61, we observed cerium peroxide precipitates under TEM, and found the most H_2_O_2_ localized on the plasma membrane of xylem parenchyma cells (Figure 6). The high electron-density precipitates first slightly increased, then decreased on plasma membrane of xylem parenchyma cells in Rb49 plant from 0 DAI to 3, 5 DAI (Figure 6A and the enlarged figure below). However, the high electron-density precipitates largely increased on plasma membrane of xylem parenchyma cells in Mudanjiang 8 plant from 0 DAI to 3, 5 DAI (Figure 6B and the enlarged figures below). The cell with more than 1 μm high electron-density plasma membrane was defined as the xylem parenchyma cell with cerium peroxide precipitates [37]. Meanwhile, the number of xylem parenchyma cells with cerium peroxide precipitates in Mudanjiang 8 at 3, 5 DAI was 5-foldas large as that in the Rb49 plant at 3, 5 DAI (Figure 6C). The lesion length of Mudanjiang 8 was larger than that of Rb49 at 20 DAI (Figure 6D).

### 2.7. Autophagy Inhibitor 3-Methyladenine Reduced Calcium Ions Accumulation in Vacuole of Mesophyll Cells in Impaired Xa3/Xa26-Mediated Resistance Reaction

Calcium ions can specifically react with potassium pyroantimonate to form calcium pyroantimonate precipitates which show high electron-density under TEM [38]. To examine the effect of autophagy inhibitor 3-MA on calcium ion distribution in resistant Rb49 and susceptible Mudanjiang 8 plants inoculated with *Xoo* strain PXO61, we analyzed the cytochemical localization of the calcium ion in rice leaves by calcium pyroantimonate precipitate method (Figure 7). At 3 DAI, calcium pyroantimonate precipitates were mainly accumulated in vacuole of mesophyll cell of Rb49 leaves infiltrated with PXO61 in H_2_O solution (Figure 7A-1), while the calcium pyroantimonate precipitates were mostly accumulated in the intercellular gaps of mesophyll cells of Rb49 leaves infiltrated with PXO61 in 3-MA solution (Figure 7A-2). However, little calcium pyroantimonate precipitates were accumulated in vacuole and intercellular gaps of mesophyll cells of Rb49 leaves infiltrated with only 3-MA solution (Figure 7A-3). Meanwhile, there were a large number of calcium pyroantimonate precipitates in the intercellular gaps of mesophyll cell and few precipitates in vacuole of mesophyll cell of Mudanjiang 8 plant leaves infiltrated with PXO61 in H_2_O solution at 3 DAI (Figure 7B-4). The number of mesophyll cells with calcium pyroantimonate precipitates in vacuole of Rb49 plant leaves infiltrated with PXO61 in H_2_O solution was significantly (*p* < 0.01) larger than the number of mesophyll cells treated in 3-MA solution (Figure 7C). It was also larger than the number of mesophyll cells infiltrated with only 3-MA solution and the number of mesophyll cells in Mudanjiang 8 plant infiltrated with PXO61 in H_2_O solution (Figure 7C). At the same time, the numbers of calcium pyroantimonate precipitates in the intercellular gaps between mesophyll cells of Rb49 plant leaves infiltrated with PXO61 in 3-MA solution and of Mudanjiang 8 plant leaves infiltrated with PXO61 in H_2_O solution were larger than that of the Rb49 plant leaves infiltrated with PXO61 in H_2_O solution or only 3-MA solution (Figure 7D). The number of inoculation sites with water-soaked symptoms in Rb49 plant leaves infiltrated with PXO61 in 3-MA solution was higher than that of Rb49 plant leaves infiltrated with PXO61 in H_2_O solution (Appendix A).

### 2.8. Accumulation of Calcium Ions in Vacuole of Xylem Parenchyma Cells in Xa3/Xa26-Mediated Resistance

Ultrastructural analysis also showed that resistant Rb49 plant and susceptible Mudanjiang 8 plant leaves inoculated with *Xoo* strain PXO61 exhibited different accumulation of calcium pyroantimonate precipitates (high electron-density) in xylem vessel and xylem parenchyma cells (Figure 8). In Rb49 plant leaves at 3, 5 DAI, there was a large amount of high electron-density precipitates in vacuole of xylem parenchyma cell, and no marked change of precipitates in xylem vessel was observed (Figure 8A). At 0 DAI, few precipitates were found to be accumulated in the vacuole and xylem vessel (Figure 8A). However, in Mudanjiang 8 plant leaf at 3, 5 DAI, a large number of high-electron density precipitates accumulated in the xylem vessel and on the plasma membrane of xylem parenchyma cell, and few precipitates were accumulated in vacuole of xylem parenchyma cell, or on plasma membrane, or in xylem vessel at 0 DAI (Figure 8B). Furthermore, the number of xylem parenchyma cells with calcium pyroantimonate precipitates in vacuole in Rb49 plant was 5 folds as large as the number of xylem parenchyma cells in Mudanjiang 8 plant at 3, 5 DAI (Figure 8C). Meanwhile, the number of xylem parenchyma cells with calcium pyroantimonate precipitates on plasma membrane in Rb49 plant was significantly (*p* < 0.01) smaller than that of the Mudanjiang 8 plant at 5 DAI (Figure 8D). For Mudanjiang 8 plant, the number of xylem parenchyma cells with precipitates on plasma membrane was markedly higher (*p* < 0.05 and *p* < 0.01) at 3, 5 DAI than at 0 DAI (Figure 8D). At the same time, the number of precipitates in xylem vessel of Rb49 plant was smaller than that of the Mudanjiang 8 plant at 3, 5 DAI, and the number of calcium pyroantimonate precipitates in xylem vessel of Mudanjiang 8 plant was significantly larger (*p* < 0.05 and *p* < 0.01) at 3, 5 DAI than at 0 DAI (Figure 8E). The lesion length of Mudanjiang 8 was larger than that of the Rb49 at 14 DAI (Figure 8F).

## 3. Discussion

### 3.1. Xa3/Xa26 Induces Autophagy-Like Cell Death to Defend Xoo

In *Arabidopsis thaliana*, many autophagy related-genes (*ATG*) are involved in plants resistance to pathogens [12,14,39]. *Arabidopsis ATG5* and *ATG7* trigger resistance to necrotrophic pathogen *Alternaria brassicicola* and confer oxidative stress [40]. *ATG6* plays an essential role in *Arabidopsis* resistance to bacterial pathogen *Pseudomona syringae* pv. tomato (*Pst*) [12,13]. In rice, some autophagy related-genes (*OsATG*) display differential expression patterns in the resistant or susceptible reactions to fungal pathogen *Magnaporthe grisea* and bacterial pathogen *Xoo* [31]. Furthermore, the *MR* genes *Xa1*, *Xa4,* and *Xa21* mediated resistance to *Xoo* are associated with autophagy-like cell death characterized by formation of autophagosome-like bodies in xylem parenchyma cells [9]. Meanwhile, the *OsATG5* and *OsATG7* are induced to higher levels in *Xa1*-, *Xa4*- and *Xa21-*mediated resistance to *Xoo* than they are in susceptible reaction [9]. In this study, we found that the *OsATG*s were induced to a significantly higher level in *Xa3/Xa26-*mediated resistance than they were in susceptible reaction (Figure 2 and Appendix A). In addition, autophagosome-like bodies were observed in xylem parenchyma cells and mesophyll cells of *Xa3/Xa26-*containing rice plants infected by *Xoo* (Figure 1, Appendix A). Taken together, the present results indicate that autophagy-like cell death occurs in *Xa3/Xa26-*mediated resistance.

Although autophagy is dispensable for *Arabidopsis* resistance to biotrophic bacterium *Pst* with avirulent genes, and it merely is a HR-PCD pathway [14], autophagy controlled by *ATG6* kept healthy uninfected cells alive by restricting HR-PCD, thus autophagy is a pre-survival pathway in maintaining *Arabidopsis* resistance to *Pst* [12,13]. Autophagy can be inhibited by 3-methyladenine (3-MA) in plant cells [33,41,42]. In rice resistance to *Xoo*, 3-MA partially inhibits *Xa1*-, *Xa4*-*,* and *Xa21*-mediated autophagy-like cell death [9]. Such cell death in turn keeps xylem parenchyma alive to facilitate rice resistance to *Xoo* [9]. The 3-MA also partially impaired *Xa3/Xa26-*mediated resistance by reducing the formation of autophagosome-like bodies in mesophyll cells (Figure 3). These results indicate that the autophagy-like cell death is partially involved in the *Xa3/Xa26-*mediated resistance. *Xa3*/*Xa26* is similar to *Xa21* in encoding LRR receptor and transferring resistance signals into rice cells [9]. The autophagy-like cell death triggered by *Xa3*/*Xa26* also could transfer signals to keep xylem parenchyma cells intact, in contrast to the protoplast shrinkage of xylem parenchyma cells accompanied with plasma membrane rupture in susceptible reaction to *Xoo* (Figure 1, Appendix A). Therefore, the autophagy-like cell death may provide a vital cell environment to facilitate *Xa3*/*Xa26-*mediated resistance against *Xoo*.

### 3.2. Autophagy-Like Cell Death Negatively Regulates H_2_O_2_ Accumulation to Reduce Oxidative stress in Xa3/Xa26-Mediated Resistance

In *Arabidopsis* resistance to bacterial pathogen *Pst* with avirulent gene *avrRpm1*, autophagy negatively regulates NPR1-dependent salicylic acid (SA) signaling and partially suppresses H_2_O_2_ accumulation in SA-independent pathway [43]. *NH1*, the rice ortholog of *Arabidopsis NPR1*, is rapidly induced to reach a higher expression level in *Xa3/Xa26*-mediated reaction than in the susceptible reaction [27]. This is, except at 0.5, 1 h after inoculation (HAI) with *Xoo* strain PXO61, when the SA concentration of susceptible rice leaves to *Xoo* is always higher than the SA concentration in resistant Minghui 63 plant leaves with *Xa3*/*Xa26* at 2, 12, 24, 72 and 168 HAIs [44]. High SA content induces chlorotic cell death at non-inoculation sites of *Arabidopsis ATG2* and *ATG5* mutants infected by *Pst* with avirulent gene avrRpm1 [43]. At 3 DAI, the high SA content in susceptible reaction to *Xoo* [44] might have caused the chlorotic and wilt lesion along the leaf inoculation sites (data non-shown). At 3 DAI, a large number of the autophagy-like bodies occurred, and some representative *OsATG*s were remarkably higher expressed in *Xa3*/*Xa26*-mediated resistance plants than in the susceptible plants (Figure 1, Figure 2 and Appendix A). In *Xa3*/*Xa26*-mediated resistance, there were more H_2_O_2_ accumulation on inoculation sites when autophagy-like cell death was inhibited by 3-MA than when not (Figure 4). The H_2_O_2_ concentration in susceptible rice leaves was always higher than that in resistant rice leaves with *Xa3*/*Xa26* (Figure 5). Meanwhile, the cerium peroxide precipitation analysis also indicated that the number of xylem parenchyma cells with H_2_O_2_ in susceptible reaction was 5-fold that of the *Xa3*/*Xa26*-mediated resistance (Figure 6). These results suggest that autophagy-like cell death negatively regulates H_2_O_2_ accumulation, which may be independent of the SA signal pathway in *Xa3/Xa26*-mediated resistance reaction.

H_2_O_2_ is the major and most stable type among ROS in plant oxidative burst [15], but the high concentration of H_2_O_2_ may damage plasma membrane, resulting in cell death [15]. A large amount of H_2_O_2_ accumulation on the plasma membrane of xylem parenchyma cells was accompanied with protoplast shrinkage and plasma membrane rupture in rice leaves susceptible to *Xoo* at 3, 5 DAI (Figure 1, Figure 6 and Appendix A). Although the H_2_O_2_ signal pathway can be activated by XA3/XA26 interacting with OsTPI1.1 in *Xa3/Xa26*-mediated resistance reaction [22], the H_2_O_2_ continuous accumulation on plasma membrane could damage the integrity of plasma membrane. In inoculation sites, the H_2_O_2_ content in resistant rice leaves with *Xa3*/*Xa26* was lower than that in susceptible rice leaves at 3, 5 DAI (Figure 5A). Therefore, the protoplast shrinkage of xylem parenchyma cell accompanied by plasma membrane rupture (Figure 1B, Appendix A) is possibly caused by the high H_2_O_2_ accumulation on plasma membrane in rice susceptible reaction to *Xoo* at 5 DAI. The *Xa3*/*Xa26*-mediated resistance was partially impaired by in vitro H_2_O_2_ spraying treatment (Figure 5B). This result implies that high concentration of H_2_O_2_ has possibly caused oxidative stress to rice leaf. Under oxidative stress conditions, *AtATG18a* and *OsATG10b* are all involved in degrading oxidized proteins and protecting the survival of cells against oxidative stress [30,45]. These evidences suggest that the autophagy-like cell death negatively regulates H_2_O_2_ accumulation to presumably abolish oxidative stress in *Xa3*/*Xa26*-mediated resistance.

### 3.3. Autophagy-Like Cell Death Regulates Calcium Ion Distribution in Xa3/Xa26 Mediated Resistance

XA10, which encodes by rice *MR* gene *Xa10* and is localized on endoplasmic reticulum (ER), induces the release of ER calcium ions to elevate the concentration of cytosolic calcium ion in Hela cells [21]. Although ER, mitochondria, and chloroplasts can be an internal calcium ion source [46], the apoplast (mainly including cell wall) and the vacuole containing mM calcium ions are the main source of calcium ion in plant cells [47,48] in contrast to the cytoplasm containing calcium ions at μM concentration level [46,49]. The plant defense responses including PTI and ETI can all trigger the increase of cytosolic calcium ions by activating ion channel proteins on plasma membrane to pump the extracellular (apoplast) calcium ions into cytosol [46,50,51,52]. In *Xa3/Xa26*-mediated resistance, calcium ions gradually increased in vacuole of xylem parenchyma cells, and then decreased in xylem vessel (apopolast) next to the xylem parenchyma cells (Figure 8). But in susceptible reaction, the calcium ions gradually increased both in xylem vessel and on the plasma membrane of xylem parenchyma cells (Figure 8). These results suggest that the extracellular calcium ions can be transferred into the vacuole of xylem parenchyma cells in *Xa3/Xa26*-mediated resistance reaction, while they cannot in the susceptible reaction. No calcium pyroantimonate precipitate was detected in the cytoplasm of xylem parenchyma cell during *Xa3*/*Xa26-*mediated resistance (Figure 7A, Figure 8A), which may be because the calcium ions at low concentration level could not be detected by the calcium pyroantimonate precipitate method, or because calcium ions bound to calmodulin proteins to form non-free calcium ion [46,53]. Therefore, when the calcium ions are transferred into the vacuole, the calcium ion concentration of cytoplasm in xylem parenchyma cell is increased in *Xa3*/*Xa26*-mediated resistance. Taken together, the autophagy-like cell death regulates the distribution of calcium ions in the xylem parenchyma cell during *Xa3*/*Xa26*-mediated resistance.

The *MR* gene *Xa10* also increases calcium ion concentration of cytosol to induce HR-PCD in rice resistance to *Xoo* [21]. The HR-PCD induced by *Xa10* is characterized by apoptosis in Hela cells [21]. However, the HR-PCD induced by *Xa3/Xa26* belonged to autophagy-like cell death which was inhibited by autophagy inhibitor 3-MA (Figure 1, Figure 2 and Figure 3). The 3-MA blocked the extracellular calcium ion transferring through cytoplasm to vacuole of mesophyll cell in *Xa3/Xa26*-mediated resistance (Figure 7). The above evidences suggest that the autophagy-like cell death may activate calcium ion signals to participate in *Xa3/Xa26* resistance to *Xoo*.

## 4. Materials and Methods

### 4.1. Rice Materials

We used two pairs of resistant and susceptible rice lines: Rb49 and Mudanjiang8, IRBB3 and IR24. Rb49 and IRBB3 all carry *Xa3/Xa26* and exhibit race-specific resistance against *Xoo,* whereas Mudanjiang 8 and IR24 are susceptible to *Xoo* [23,25]. IRBB3 and IR24 are *Oryza sativa indica* near-isogenic lines. Transgenic line Rb49 carries *Xa3/Xa26* driven by its native promoter, and it has the genetic background of *Oryza sativa japonica* cultivar Mudanjiang 8 [23].

### 4.2. Pathogen Inoculation

For bacterial inoculum preparation, Philippine *Xoo* strain PXO61 was seeded on a potato semisynthetic agar medium and incubated at 28 °C for 3 days [54]. Inoculum was prepared by suspending the bacterial culture with sterilized water to a concentration of about 10^9^ cells ml^−1^, measured by nephelometry method [55].

For the analysis of the cell death ultrastructure and autophagy-related gene expressions, IRBB3, IR24 at tilling stage and Rb49, Mudanjiang 8 at four-leaf stage were inoculated with Philippine *Xoo* strain PXO61 by the leaf-clipping method [54]. At tilling stage of IRBB3 and IR24 (approximately 40 days after transplanting), 6 to 10 of the uppermost fully expanded leaves of each plant were cut with a pair of scissors predipped in the bacterial suspension. At the four-leaf stage of Rb49 and Mudanjiang 8 (approximately 20 days after transplanting), 2 to 3 of the fully expanded leaves of each seedling were cut with scissors predipped in bacterial suspension. The about 2 mm and 3 cm long leaf fragments adjacent to the inoculate site were used for ultrastructure and gene expression analysis respectively. The extent of disease was scored by measuring the lesion length (cm) or lesion area (lesion length/leaf length, %) at 2 weeks or 20 days after inoculation.

To analyze the mesophyll cell ultrastructure and the influences of autophagy inhibitor 3-MA on resistance, rice plants were inoculated with 10^9^ cells ml^-1^
*Xoo* strain PXO61 respectively in H_2_O solution, in 5 mM 3-MA (Sigma, SIGMA-ALDRICH, MS, USA) solution, and only 5 mM 3-MA solution without bacterium by infiltrating leaf using the needleless syringe [56]. The suspension was prepared by nephelometry in each solution to determine bacterium concentration [55]. During Rb49 and Mudanjiang 8 plants four-leaf stages, different suspensions were infiltrated into leaves from the underside by pressing the mouth of syringe to the leaf, then the syringe was supported with a finger propped behind it and gently depressing the plunger. Along the line perpendicular to the leaf midrib, two sites were infiltrated on the left and right sides of leaf midrib. Total six sites were infiltrated on each leaf. Each two sites on the left and right sides of midrib spaced about 0.5 cm. The extent of disease was evaluated by accounting the number of infiltration site with water-soaked symptom on day 3 after inoculation [57].

All the inoculation of plants with *Xoo* was biologically repeated at least twice with similar results, and one replicate was shown.

### 4.3. Cell Death Ultrastructure Analysis

The ultrastructure of cell death was studied by transmission electron microscopy (TEM). The leaf fragments of inoculation sites were cut into about 2 mm^2^ small pieces using a sharp scissor. Immediately, the leaf pieces were fixed in 2.5% (*w*/*v*) glutaraldehyde in 0.1 mol/L phosphate buffer solution (PBS) (pH 7.2) overnight at 4 °C Then, the leaf pieces were washed in PBS three times at room temperature (20–25 °C) for 30 min each time, and post-fixed for 2 h in 1% osmium tetroxide. Subsequently, the leaf pieces were rewashed in PBS three times, dehydrated in graded series of acetone, infiltrated with Spurr resin (SPI, SPI Chem, PA, USA) and polymerized at 60 °C for 48 h. The specimens were cut into ultra-thin sections (60–70 nm thickness), mounted on copper grid, stained with 2% uranyl acetate and observed by a transmission microscope (H-7650; Hitachi, Tokyo, Japan) at 80 kv. Images were captured by a CCD camera (Model 832 ORIUS, Gatan, PA, USA). At least 9 ultra-thin sections from 3 independent rice leaves were examined by TEM. Every cell death ultrastructure analysis was biologically repeated at least twice with similar results, and one replicate was shown.

### 4.4. Autophagy-Related Gene Expression Analysis

Total RNA was isolated from the 3-cm leaf fragments next to inoculation sites. Total RNA was extracted by using Trizol reagent (Invitrogen, Carlsbad, California, USA). An aliquot (5 µg) of total RNA was treated with RNase-free DNase I (Invitrogen) to remove potentially contaminating DNA, and first-strand cDNA was reverse transcribed from total RNA with oligo(dT)_18_ primer using M-MLV reverse transcriptase (Promega, Madison, WI, USA) according to the manufacturer’s instructions. Quantitative reverse transcription polymerase chain reaction (qRT-PCR) was conducted by autophagy-related gene specific primers (Appendix A), as described previously [58]. The expression level of the rice actin gene was used to standardize the RNA sample of each RT-PCR, and the expression level relative to that of IR24 leaf at 0 h after inoculation was calculated. Each qRT-PCR assay was biologically repeated twice with similar results, with each repetition having three technical replicates and only one biological replicate was presented.

### 4.5. DAB Staining, H_2_O_2_ Treatment, and H_2_O_2_ Content Assay

To detect H_2_O_2_ accumulation in the rice leaves infected to *Xoo*, the detached leaves at inoculation sites were immediately immersed into 1% (*w*/*v*) 3,3′-diaminobenzidine (DAB) (Sangon, Sangno Biotech) in 10 mmol/L 2-(N-morpholino) ethanesulfonic acid (Sangon, Sangno Biotech, Shanghai, China) buffer (pH 6.5), and then were vacuum-infiltrated for 30 min. Afterwards, the leaves were incubated at room temperature for 2 days in absence of light. Stained leaves were bleached by ethanol until a clear background was obtained [59]. The stained leaves were photographed by scientific scanner (Image Scanner III, GE, Sweden). To semi-quantify the intensity of H_2_O_2_ accumulation at infiltrating inoculation sites, the digitized images of stained inoculation site were analyzed by ImageJ software (version 1.47 t, NIH, WA, USA). DAB color of each inoculation sites was processed by Colour Deconvolution plugin 1.0, then was measured to obtain the mean gray value. The 255 divided by mean gray value was used to indicate the intensity of H_2_O_2_ accumulation [60].

For evaluating the influence of in vitro H_2_O_2_ on the rice resistance, the inoculated leaves were sprayed by 25 mM H_2_O_2_ solution lasting for 5 min on 8:00 am, 12:00 am, and 16:00 pm every day from day 1 to day 14 after leaf-clipping inoculation.

Quantitative measurement of H_2_O_2_ content was performed using the Amplex Red hydrogen peroxide/peroxidase assay kit (A22188, Molecular Probes, ThermoFisher SCIENTIFIC, CA, USA) according to the manufacturer’s instruction. Briefly, 3-cm long detached leaf fragments next to leaf-clipping inoculation site were immediately immersed into liquid nitrogen and stored in −70 °C. H_2_O_2_ was extracted from leaves and measured in a 96-well plate by luminescence spectrometer (Infinite M200, Tecan, Switzerland) according to the method described previously [61].

The DAB staining, H_2_O_2_ treatment, and H_2_O_2_ content assay were biologically repeated at least twice with similar results, and one replicate was shown.

### 4.6. H_2_O_2_ and Calcium Ion Localization Assay

Cytochemical detection of H_2_O_2_ was carried out by the modified method described previously [37]. Briefly, leaf pieces (~1 to 2 mm^2^) were excised from inoculation leaves and incubated in freshly prepared 5 mmol/L CeCl_3_ in 50 mmol/L 3-(N-morpholino) propanesulfonic acid (pH 7.2) for 1 h. Samples were then pre-fixed in 1.25% (*v*/*v*) glutaraldehyde /1.25% (*v*/*v*) paraformaldehyde in 50 mM sodium cacodylate (CAB) buffer (pH 7.2) for 4 h. Then, samples were washed three times for 30 min in CAB buffer and post-fixed for 2 h in 1% (*w*/*v*) osmium teroxide in CAB buffer. Samples were then washed three times for 30 min. Subsequently, the samples were dehydrated, infiltrated, embedded, and observed, following the methods described in above ultrastructure analysis.

Subcellular localization of calcium ion was performed by the method described by Yin et al. (2014) with minor modification. The inoculated leaves were cut into about 2 mm^2^ slices and immediately fixed in 2.5% (*v*/*v*) glutaraldehyde / 2% (*v*/*v*) paraformaldehyde in 2% (*w*/*v*) potassium pyroantimonate 60 mmol/L potassium phosphate buffer (pH 7.8) at 4 °C for 12 h. Then, the samples were washed three times in wash buffer containing 2% (*w*/*v*) potassium pyroantimonate in 60 mmol/L potassium phosphate buffer (pH 7.8) for 30 min each time. Afterwards, the samples were post-fixed in containing 1% (*w*/*v)* osmium tetroxide dissolved in wash buffer at room temperature for 2 h, and washed three times for 30 min each time in wash buffer. Subsequently, the samples were dehydrated, infiltrated, embedded, and observed following the methods described in above ultrastructure analysis.

The H_2_O_2_ localization and calcium ion localization assays were biologically repeated at least twice with similar results, and one replicate was shown.

### 4.7. Statistical Analysis

The statistical analyses of significant differences between resistance plants and susceptible plants, between treated plants and control plants were conducted by pairwise Student *t*-tests in EXCEL (Microsoft, http://www.microsoftstore.com).

## 5. Conclusions

In *Xa3/Xa26* mediated resistance, autophagy-like cell death occurs in xylem parenchyma cells to eliminate H_2_O_2_ accumulation on plasma membrane accompanied by the extracellular calcium ion transferring into the vacuole. The autophagy-like cell death induced by *Xa3/Xa26* may abolish the oxidative stress and possibly activates calcium ion signals to participate in *Xa3/Xa26*-mediated resistance.

## Figures and Tables

**Figure 1 ijms-21-00194-f001:**
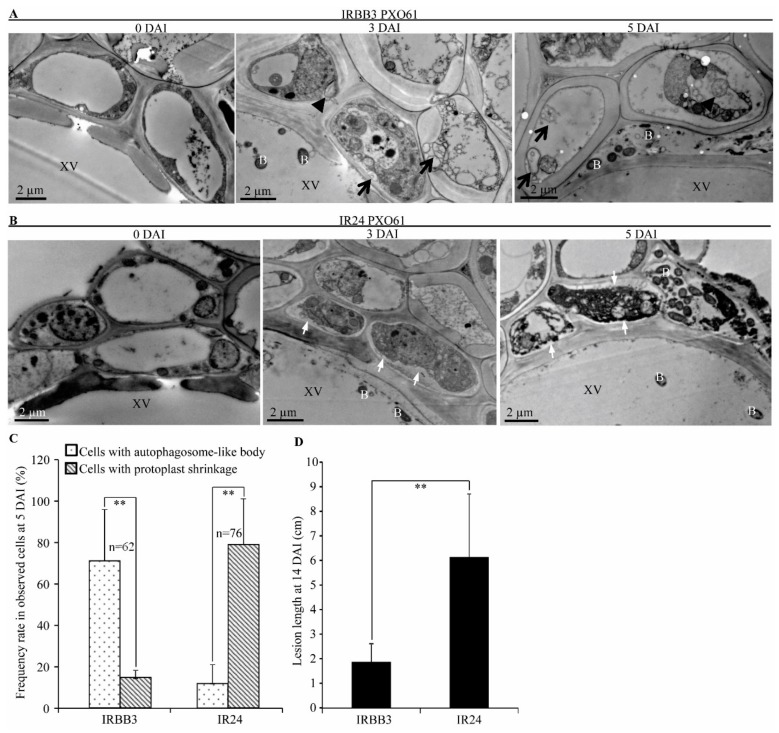
Autophagy-like ultrastructure features of xylem parenchyma cells in *Xa3/Xa26* mediated-resistance in comparison with protoplast shrinkage in susceptible reaction. XV, xylem vessel; B, *Xoo* bacteria; black arrow, autophagosome-like bodies with double membrane; black arrowhead, autophagosome-like body with single membrane; white arrow, protoplast shrinkage. (**A**) Autophagosome-like bodies in xylem parenchyma cells in IRBB3 plants infected with *Xoo* strain PXO61 at Day 3 and 5 after inoculation (DAI) in comparison with 0 DAI. (**B**) Protoplast shrinkage of xylem parenchyma cells in IR24 plants inoculated with *Xoo* strain PXO61 at 3 and 5 DAI, in comparison with 0 DAI. (**C**) Percentage of cells with autophagosome-like bodies and protoplast shrinkage in xylem parenchyma cells at 5 DAI. Data are presented as mean (62–76 xylem parenchyma cells of three xylem vessels from three different plants) + standard deviation (SD). Double asterisks (**) stand for a significant difference between frequency rate of cells with autophagosome-like body and frequency rate of cells with protoplast shrinkage at *p* < 0.01; n, the number of total observed cells. (**D**) Leaf lesion length of IRBB3 and IR24 challenged with *Xoo* at 14 DAI. Bars represent mean (10–20 leaves from four plants) + SD. Double asterisks (**) stand for the significant difference between resistant and susceptible plant at *p* < 0.01.

**Figure 2 ijms-21-00194-f002:**
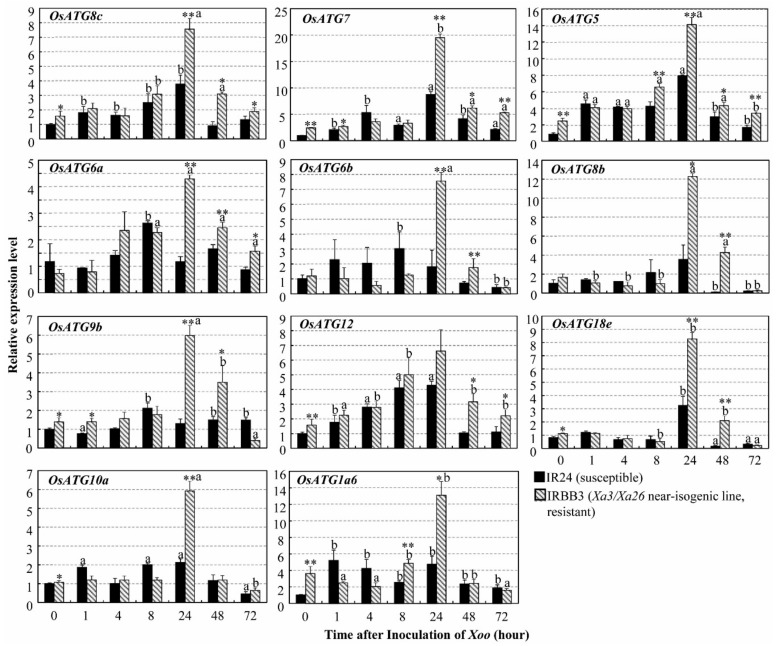
Autophagy-related gene (*ATG*) expression level in *Xa3/Xa26-*mediated resistance. Expression of autophagy-related genes analyzed by qRT-PCR in resistant (IRBB3) and susceptible (IR24) rice plants. Rice plants were inoculated with *Xoo* strain PXO61 at tilling stage. Data are presented as mean (three replicates) + standard deviation (SD). Letters “a” and “b” indicate statistically significant differences between 0 h and other time at *p* < 0.01 and *p* < 0.05, respectively in same rice plants. Double asterisks (***p* < 0.01) and single asterisk (**p* < 0.05) indicate different levels of statistical significance between IRBB3 and IR24 inoculated at the same time point.

**Figure 3 ijms-21-00194-f003:**
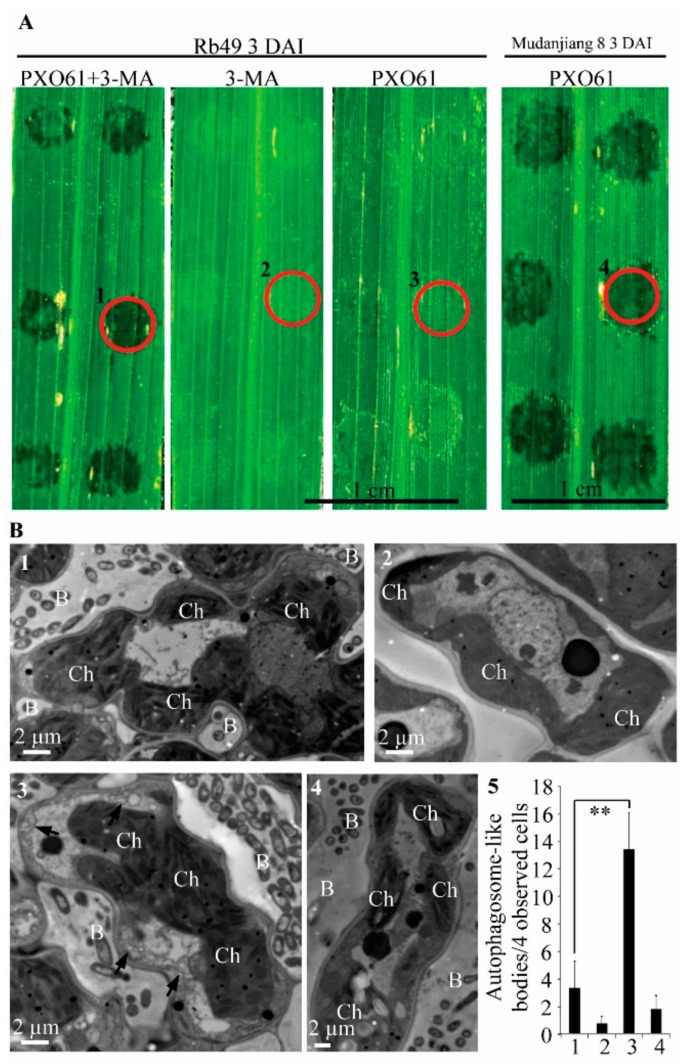
Effect of autophagy inhibitor 3-methyladenine on the water-soaked symptoms and autophagosome-like body number at inoculation sites. (**A**) The symptom of infiltration sites on Rb49, Mudanjiang 8 leaves inoculated with *Xoo* strain PXO61 in H_2_O solution, in 3-methyladenine (3-MA) solution, and with only 3-MA solution. The 1, 2, 3 and 4 red circles represent the observed areas corresponding to the numbers of Figure B. (**B**) Number of autophagosome-like bodies reduced by 3-MA in mesophyll cell during *Xa3/Xa26*-mediated resistance at 3 DAI. B, *Xoo* bacterium; Ch, Chloroplast. (1) Few autophagosome-like bodies in Rb49 leaf infiltrated with PXO61 in 3-MA solution. (2) No autophagosome-like bodies in Rb49 leaf infiltrated with only 3-MA solution. (3) Many autophagosome-like bodies (black arrow) in Rb49 leaf infiltrated with PXO61 in H_2_O solution. (4) Few autophagosome-like bodies in Mudanjiang 8 leaf infiltrated with PXO61 in H_2_O solution. (5) Number of autophagosome-like bodies per 4 observed mesophyll cells in Rb49 leaf infiltrated 1) with PXO61 in 3-MA solution, 2) with only 3-MA solution, 3) with PXO61 in H_2_O solution, and 4) in Mudanjiang 8 leaf infiltrated with PXO61 in H_2_O solution. Data represent mean (at least 3 independent leaf areas of 4 observed cells from 3 plants) + standard deviation. Double asterisks (**) indicate a significant difference (*p* < 0.01) between 1) Rb49 infiltrated with PXO61 in H_2_O solution and 3) Rb49 infiltrated with PXO61 in 3-MA solution.

**Figure 4 ijms-21-00194-f004:**
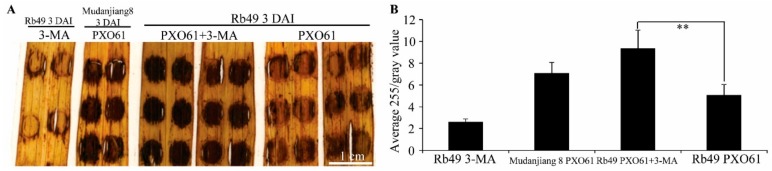
H_2_O_2_ accumulation promoted by autophagy inhibitor 3-methyladenine in impaired *Xa3*/*Xa26-*mediated resistance reaction. (**A**) DAB staining of Rb49 and Mudanjiang 8 leaves with inoculation sites infiltrated by *Xoo* strain PXO61 in H_2_O solution, in 3-methyladenine (3-MA) solution, and infiltrated with only 3-MA solution at 3 DAI. (**B**) Average H_2_O_2_ content of inoculation site showed by 255 divided by gray value of inoculation site. Data represent mean (18-24 inoculation sites of 3 independent plants) + standard deviation. Double asterisks (**) indicate H_2_O_2_ content statistically significant differences (*p* < 0.01) between Rb49 infiltrated with PXO61 in 3-MA solution and that in H_2_O solution.

**Figure 5 ijms-21-00194-f005:**
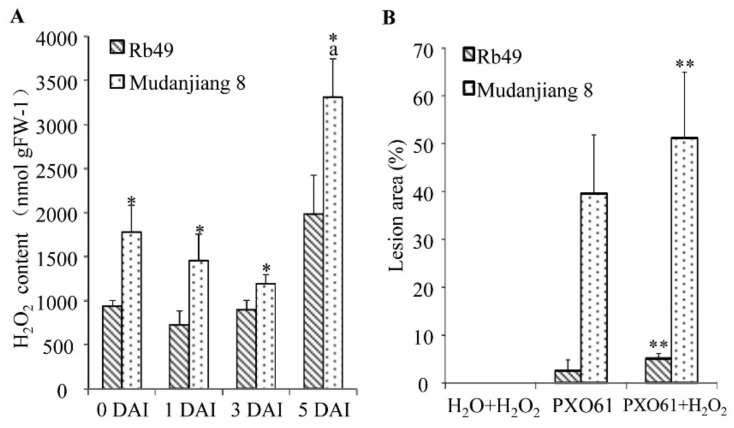
Small amounts of H_2_O_2_ accumulation in *Xa3/Xa26-*mediated resistance and the increased rice susceptibility to *Xoo* by in vitro H_2_O_2_ application. (**A**) H_2_O_2_ content of Rb49 and Mudanjiang 8 leaves infected by *Xoo* strain PXO61. Data represent the mean (total of three leaves from three plants) + standard deviation (SD). Letter “a” indicates statistically significant differences between 0 day and other time in the same rice plant (*p* < 0.01). Double asterisks (** *p* < 0.01) and single asterisk (* *p* < 0.05) indicate statistically significant differences between Rb49 and Mudanjiang 8 at same inoculation time point. (**B**) Effect of in vitro H_2_O_2_ spraying on lesion area of Rb49 and Mudanjiang 8 leaves inoculated by H_2_O and inoculated by *Xoo* strain PXO61. Double asterisks (**) indicate significant differences (*p* < 0.01) between sprayed H_2_O_2_ plants and unsprayed H_2_O_2_ plants after inoculation with PXO61. Each bar represents mean (11–24 leaves from three plants) + SD.

**Figure 6 ijms-21-00194-f006:**
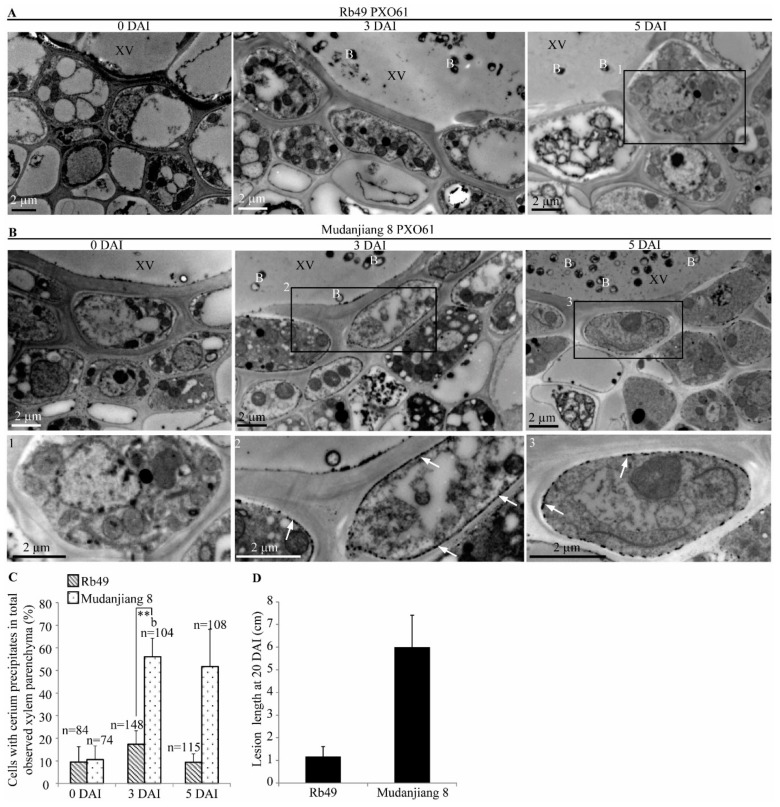
Subcellular localization of H_2_O_2_ in xylem parenchyma cell in *Xa3/Xa26-*mediated resistance. XV, xylem vessel; B, *Xoo* bacterium; White arrow, high electron-density cerium peroxide precipitates. (**A**) Small number of high electron-density cerium peroxide precipitates (H_2_O_2_) on plasma membrane of xylem parenchyma cells in Rb49 plants inoculated by *Xoo* strain PXO61 at 0, 3, 5 DAI. In Figure A, 1 represents the enlarged xylem parenchyma cells in the 1st rectangle. (**B**) Large number of high electron-density cerium peroxide precipitates (H_2_O_2_) on plasma membrane of xylem parenchyma cells in Mudanjiang 8 plants infected by PXO61 at 3, 5 DAI, in contrast with no cerium peroxide precipitates at 0 DAI. In Fig.B, 2, 3 represents the enlarged xylem parenchyma cells in the 2nd, 3rd rectangle. (**C**) Percentage of cells with cerium peroxide precipitates on plasma membrane in all the observed xylem parenchyma cells. Data represent mean (74-115 xylem parenchyma cells of three xylem vessels from three different plants) + standard deviation (SD). Double asterisks (**) indicate significant difference (*p* < 0.01) in the number of xylem parenchyma cell between Rb49 and Mudanjiang 8 at 3 DAI. Letter “b” indicates statistically significant difference between 0 day and other time in the same rice plant at *p* < 0.05. n, the number of all the observed cells. (**D**) Lesion length of Rb49 and Mudanjiang 8 leaves infected by *Xoo* at 20 DAI. Each bar represents mean (11–15 leaves from four plants) + SD.

**Figure 7 ijms-21-00194-f007:**
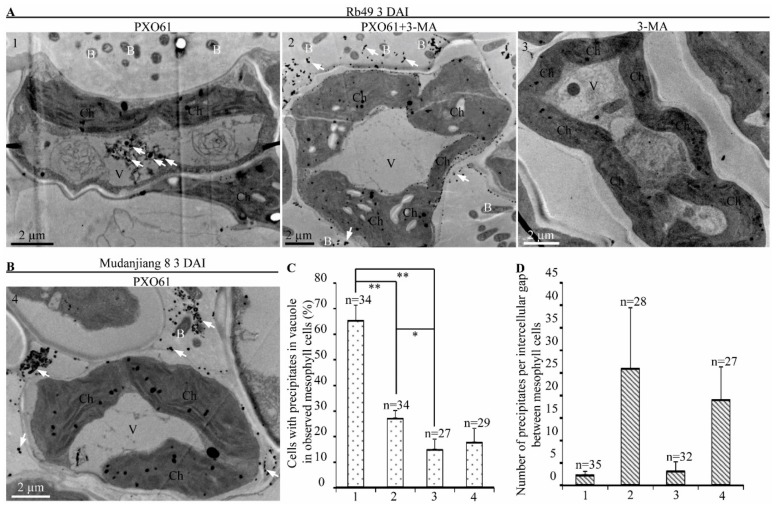
Calcium ion distribution in mesophyll cells affected by autophagy inhibitor 3-methyladenine at 3 DAI. B, *Xoo* bacterium; Ch, chloroplast; V, vacuole; white arrow, calcium pyroantimonate precipitates (high electron-density). (**A**) 3-methyladenine (3-MA) changed the distribution of calcium pyroantimonate precipitates in Rb49 plants infiltrated with *Xoo* strain PXO61 in H_2_O solution, in 3-MA solution, and with only 3-MA solution. (**B**) Calcium precipitates were mainly localized in the intercellular gaps of mesophyll cells in Mudanjiang 8 infiltrated with PXO61 in H_2_O solution. (**C**,**D**) Percentage of cells with calcium precipitates in vacuole in all the observed mesophyll cells and number of calcium precipitates per intercellular gap between mesophyll cells. Data represent mean (27–34 mesophyll cells from 3 plant leaves and 27–35 precipitates distributed in intercellular gaps between mesophyll cells from 3 plant leaves) + standard deviation. n, the number of all the observed cells and the number of all the observed intercellular gaps. Double asterisks (**) indicate the significant differences between PXO61 infiltration in H_2_O solution and PXO61infiltration in 3-MA solution, between PXO61infiltration in H_2_O solution and only 3-MA solution infiltration at *p* < 0.01 in Rb49 plants. Single asterisk (*) indicates the significant difference between PXO61 infiltration in 3-MA solution and only 3-MA solution infiltration at *p* < 0.05 in Rb49 plants. The 1, 2, 3, and 4 represent Rb49 plant infiltrated by PXO61 in H_2_O solution, that by PXO61 in 3-MA solution, that infiltrated only by 3-MA solution, and Mudanjiang 8 plant infiltrated by PXO61 in H_2_O solution, respectively.

**Figure 8 ijms-21-00194-f008:**
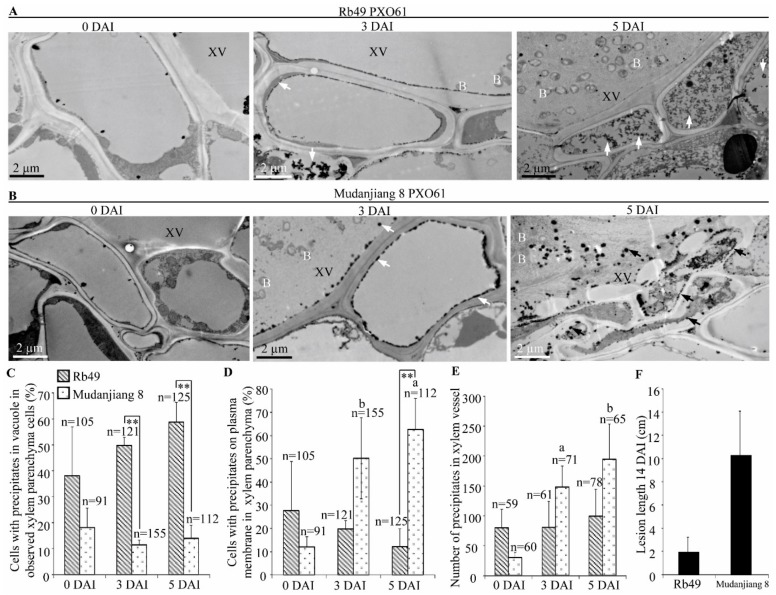
Calcium ion accumulation induced by *Xa3/Xa26* in vacuole of xylem parenchyma cell. XV, xylem vessel; B, *Xoo* bacterium; white arrow and black arrow, high-electron density calcium precipitates. (**A**) Numerous high-electron density calcium precipitates (indicated by white arrow) in vacuole of xylem parenchyma cells in Rb49 plant leaves after inoculation by *Xoo* strain PXO61 at 3, 5 DAI, in comparison with 0 DAI. (**B**) The gradually increasing calcium precipitates (indicated by white arrow and black arrow) on plasma membrane of xylem parenchyma cell and xylem vessel lumen in Mudanjiang 8 plant leaves at 3, 5 DAI in contrast to 0 DAI. (**C**,**D**,**E**) Percentage of cells with calcium precipitates in vacuole (**C**) or on plasma membrane (**D**) of xylem parenchyma cells in all the observed cells, and the number of calcium precipitates in xylem vessel (**E**). Data represent the mean (91–155 xylem parenchyma cell of three xylem veins and 59–78 calcium precipitates of three xylem vessels from three different plants) + standard deviation (SD); n, the number of all the observed xylem parenchyma cells and the number of all the observed calcium precipitates in xylem vessels. Double asterisks (** *p* < 0.01) indicate statistically significant differences between Rb49 and Mudanjiang 8 at same inoculation time. The letters “a” and “b” indicate statistically significant differences between 0 day and 3, 5 day in the same rice plant at *p* < 0.01 and *p* < 0.05, respectively. (**F**) Lesion length of Rb49 and Mudanjiang 8 leaves infected by *Xoo* at 14 DAI. Each bar represents mean (10–20 leaves from four plants) + SD.

**Table 1 ijms-21-00194-t001:** Percentage of infiltrating inoculation site with water-soaked symptoms influenced by 3-methyladenine.

Plant Inoculation after 3 days ^1^	Water-Soaked Symptoms		N ^4^ (Inoculation Sites)
	Yes (%) ^2^	No (%) ^3^	
Rb49 PXO61	4	96	51
Rb49 PXO61+3-MA	65	35	80
Rb49 3-MA	0	100	36
Mudanjiang 8 PXO61	100	0	72

**^1^** almost 10^9^
*Xoo* strain PXO61 cells in H_2_O solution, in 5 mM 3-methyladenine (3-MA) solution and in only 3-MA solution infiltrated into leaves of Rb49 and Mudanjiang 8 plants at four-leaf stage. **^2^** the percentage of inoculation sites with water-soaked symptom in total infiltrated sites. **^3^** The percentage of inoculation sites without water-soaked symptoms in total infiltrated sites. **^4^** the number of total infiltrated inoculation sites from at least 3 independent plants.

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
