# Peer review of "Autophagy-Like Cell Death Regulates Hydrogen Peroxide and Calcium Ion Distribution in *Xa3/Xa26*-Mediated Resistance to *Xanthomonas oryzae* pv. *oryzae"

_ijms, 2019, doi:10.3390/ijms21010194_

Round 1

Reviewer 1 Report

Numbers of plants inoculated and samples size have no always been reported.

In this article, the authors study the autophagy-like cell death in the response of rice a Xanthomonas.

It is a well written article and a very interested one.

The show that the inoculation with Xanthomonas Xoo strain leads to autophagy like structures in IRBB3 resistant rice cultivar but not in IR24 susceptible one. This correlates with a higher expression of autophagy related genes in inoculated IRBB3 compared to inoculated IR24.

The authors then study the effect of a autophagy inhibitor. If I understand well, they use others cultivars (Mudanjiang 8 and Rb49). It is not clear to me why this change in cultivars. Can they justify? They show that the inhibitor enhances the formation of water-soaked symptom at the inoculation sites of Rb49 leaves? They conclude at a impairment of resistance. The inhibitor is shown to indeed inhibit autophagy like structures. The authors then show that the inhibitor leads to a enhancement of H2O2 accumulation in the susceptible cultivar. They then study the calcium signaling.

It is an interesting study, on a subject that deserves high interest.

I have one concern: the reader nit familiar with rice cultivars can get lost with the cultivars. The authors should better explain the cultivars, especially when they first appear in the Result section. See for instance line 116. What id IRBB3? The authors should at least indicate it is a rice cultivar and a resistant one.

The figures should be improved.

Figure 1: panel C is very tiny. It is hard to read Y axis. May be the A and B panels should be done in column. For panel C I would suggest showing only the upper bar of the error bars. This remark is for all figures.

Figure 2: lesion area figure is too small. Indicate value in the text and put the figure in supplemental.

Figure 3/ please extract the table to make it a full table.

Figure 6: make panels C and D bigger, maybe under panels A and B.

Line 151: delete “most”

In the text you have many double spaces

Author Response

Response to Reviewer 1 Comments

Point 1: Numbers of plants inoculated and samples size have no always been reported.

Response 1: We have added the number of inoculated plants and the number of samples in all the figure legends.

major:

Point 1: The authors then study the effect of a autophagy inhibitor. If I understand well, they use others cultivars (Mudanjiang 8 and Rb49). It is not clear to me why this change in cultivars. Can they justify? They show that the inhibitor enhances the formation of water-soaked symptom at the inoculation sites of Rb49 leaves? They conclude at a impairment of resistance. The inhibitor is shown to indeed inhibit autophagy like structures. Theauthors then show that the inhibitor leads to a enhancement of H2O2 accumulation in the susceptible cultivar. They then study the calcium signaling.

Response 1: There are two pairs of resistant and susceptible cultivars used in the manucript: indica cultivars IRBB3 and IR24, japonica cultivars Rb49 and Mudanjiang 8. Xa3/Xa26-mediated resistance to Xoo is influenced by genetic background and developmental stage (Cao et al. 2007). Japonica rice plants transformed Xa3/Xa26 with its native promoter show whole-growth-stage resistance (Sun et al.2004; Cao et al. 2007). But indica rice plants carrying Xa3/Xa26 only show adult stage (from tillering stage to grain filling stage) resistance (Cao et al. 2007). The expression of Xa3/Xa26 is relative higher throughout the growth stages in japonica plants, while is gradually increased from seedling stage to adult stage in indica plants, suggesting that Xa3/Xa26-mediated resistance is influenced by by genetic background and development stage (Cao et al. 2007). We have added the description of the difference between those two pair cultivars in revised manuscript (Line 105-113).

During tillering stage, the rice leaf becomes lumpy and brittle because of many big veins formation. Most solution runs away along the surface of rice leaf veins, and the leaves are easily broken off when the rice leaves at adult stage are inoculated by infiltration method. It is very difficult to infiltrate solution into rice leaf at adult stage. So, we carried out most infiltrating inoculation on Rb49 plant at seedling stage that is resistant to Xoo and has flat and tough leaf. Furthermore, the leaf at adult stage is difficult to cut into ultrasections because of the thick cell wall of leaf cells. We have done most cellular experiments on Rb49 plant at seedling (four-leaf) stage.

Rb49 plant is a stable transgenic line and widely used in Xa3/Xa36 function research (Sun et al. 2006; Cao et al. 2007, Deng et al. 2012; Li et al. 2014; Deng et al. 2018; Liu et al. 2018). Because the transgenic process could possibly affect gene expression patterns, we mainly used the non-transgenic cultivars IRBB3 and IR24 to analyze the expression patterns of ATG genes.

Additionally, we analyzed some ATG genes expression analysis in Rb49 and Mudanjiang 8 plants in revised manuscript (supplemental figure 4). The expression patterns of some ATG genes in Rb49/Mudanjiang 8 plants were similar to that in IRBB3/IR24 plants (Line 182-184).

Point 1: I have one concern: the reader nit familiar with rice cultivars can get lost with the cultivars. The authors should better explain the cultivars, especially when they first appear in the Result section. See for instance line 116. What id IRBB3? The authors should at least indicate it is a rice cultivar and a resistant one

Response 2: We greatly appreciate your helpful suggestions. We have added detailed information of resistant and susceptible rice cultivars in Results of revised manuscript.

Point 3: The figures should be improved

Response 3: We appreciate your suggestion. We have edited all the figures in revised manuscript according to your suggestion.

Point 4: Figure 1: panel C is very tiny. It is hard to read Y axis. May be the A and B panels should be done in column. Forpanel C I would suggest showing only the upper bar of the error bars. This remark is for all figures.

Response 4: We have enlarged the panel C and D of figure 1 and only remained the upper bar of error bars in all figures in revised manuscript.

Point 5: Figure 2: lesion area figure is too small. Indicate value in the text and put the figure in supplemental.

Response 5: The lesion area figure has been enlarged and separated into supplemental figure 3 in revised manuscript.

Point 6: Figure 3/ please extract the table to make it a full table

Response 6: We have separated Figure 3 into Figure 3 and Table 1 in revised manuscript.

Point 7: Figure 6: make panels C and D bigger, maybe under panels A and B

Response 7: We have zoomed the panels C and D and put the panels C and D under panels A and B in figure 6 of revised manuscript.

Point 8: Line 151: delete “most”

Response 8: We have deleted the “most” and revised the description of ATG genes (Line 164-173).

Point 9: In the text you have many double spaces

Response 9: We appreciate your suggestion. We have deleted the space in the text of revised manuscript.

Reviewer 2 Report

The paper describes a study on the reaction of rice cultivars resistant and susceptible to Xanthomonas oryzae pv. oryzae to autophagy-like cell death and its inhibitor. The manuscript provide some novel information but it should be carefully revised before it can be considered for publication in the IJMS journal. Some specific comments were given below.

Introduction – I think that some kind of description of the rice ATG genes studied is needed, especially that some of it has been provided for A. thaliana in the Discussion. It would be beneficial for the reader to understand a story behind the study. Also, I have some doubts concerning using two pairs of resistant vs. susceptible cultivars in this study. Why it was done like this? I believe it would be better to compare the reactions of both resistant and both susceptible cultivars in ALL experiments.

The genes studied should also be introduced in the Results. The comparisons of water and 3-MA solutions of Xoo should be explained to help the reader to understand the role of the autophagy inhibitor in the study.

Combining Figure 3 with Table 1 is tricky. I recommend to separate the table and present it apart from the figure. Possibly, the description will be more clear then. Please, unify font sizes in figure descriptions. Figure 4 – why only one line (Rb49) was shown in all variants?

Section 2.5 subheading is confusing. Try to simplify it.

First paragraph of the Discussion (lines 373-387) (and other paragraphs as well) is very poorly written. There is a lot of spelling and grammar errors. Please, edit and improve it carefully.

In Materials and Methods all paragraphs are presented twice. There are also some minor errors to be corrected.

Author Response

Response to Reviewer 2 Comments

Point 1: The paper describes a study on the reaction of rice cultivars resistant and susceptible to Xanthomonas oryzae pv. oryzae to autophagy-like cell death and its inhibitor. The manuscript provide some novel information but it should be carefully revised before it can be considered for publication in the IJMS journal. Some specific comments were given below.

Response 1: Thank the reviewer for the recognition of our work. We have revised the manuscript according to your suggestions.

Point 2: Introduction – I think that some kind of description of the rice ATG genes studied is needed, especially that some of it has been provided for A. thaliana in the Discussion. It would be beneficial for the reader to understand a story behind the study.

Response 2: We have added the description about the ATG genes in revised manuscript (line 180-182).

Point 3: Also, I have some doubts concerning using two pairs of resistant vs. susceptible cultivars in this study. Why it was done like this? I believe it would be better to compare the reactions of both resistant and both susceptible cultivars in ALL experiments.

Response 3: There are two pairs of resistant and susceptible cultivars used in the manucript: indica cultivars IRBB3 and IR24, japonica cultivars Rb49 and Mudanjiang 8. Xa3/Xa26-mediated resistance to Xoo is influenced by genetic background and developmental stage (Cao et al. 2007). Japonica rice plants transformed Xa3/Xa26 with its native promoter show whole-growth-stage resistance (Sun et al.2004; Cao et al. 2007). But indica rice plants carrying Xa3/Xa26 only show adult stage (from tillering stage to grain filling stage) resistance (Cao et al. 2007). The expression of Xa3/Xa26 is relative higher throughout the growth stages in japonica plants, while is gradually increased from seedling stage to adult stage in indica plants, determines the molecular mechanism of Xa3/Xa26-mediated resistance modified by genetic background and development stage (Cao et al. 2007). We have added the description of the difference between those two pair cultivars in revised manuscript (Line 168-176).

During tillering stage, the rice leaf becomes lumpy and brittle because of many big veins formation. Most solution runs away along the surface of rice leaf veins, and the leaves are easily broken off when the rice leaves at adult stage are inoculated by infiltration method. It is very difficult to infiltrate solution into rice leaf at adult stage. So, we have carried out most infiltrating inoculation on Rb49 plant at seedling stage that is resistant to Xoo and has flat and tough leaf. Furthermore, the leaf at adult stage is difficult to cut into ultrasections because of the thick cell wall of leaf cells. We have done most cellular expriments on Rb49 plant at seedling (four-leaf) stage.

Rb49 plant is a stable transgenic line and widely used in Xa3/Xa36 function research (Sun et al. 2006; Cao et al. 2007, Deng et al. 2012; Li et al. 2014; Deng et al. 2018; Liu et al. 2018). Because the transgenic process could possibly affect gene expression patterns, we mainly used the non-transgenic cultivars IRBB3 and IR24 to analyze the expression patterns of ATG genes.

At the same time, we have added some ATG genes expression analysis in Rb49 and Mudanjiang 8 plants in revised manuscript (supplemental figure 3). The expression patterns of some ATG genes in Rb49/Mudanjiang 8 plants were similar to that in IRBB3/IR24 plants (Line 280-282).

Point 4: The genes studied should also be introduced in the Results. The comparisons of water and 3-MA solutions of Xoo should be explained to help the reader to understand the role of the autophagy inhibitor in the study.

Response 4: We greatly appreciate your helpful suggestions. we have added ATG gene introduction (Line 239-241, Line 295) and the mechanism of 3-MA inhibiting autophagy (Line 295-299) in the Results.

Point 5: Combining Figure 3 with Table 1 is tricky. I recommend to separate the table and present it apart from the figure. Possibly, the description will be more clear then. Please, unify font sizes in figure descriptions. Figure 4 – why only one line (Rb49) was shown in all variants?

Response 5: We have separated Figure 3 into Figure 3 and Table 1, then unified the font size in all the figure legends.

Rb49 plant is a stable transgenic line and widely used in Xa3/Xa36 function research (Sun et al. 2006; Cao et al. 2007, Deng et al. 2012; Li et al. 2014; Deng et al. 2018; Liu et al. 2018). Rb49 carries a single copy of transgene Xa3/Xa26 driven by its native promoter with genetic background of Mudanjiang 8, and also has morphology and growth period identical to Mudanjiang 8 (Sun et al. 2004; Deng et al. 2018; Liu et al. 2018). Rb49 and Mudanjiang 8 can be considered as near-isogenic lines, so we only used one Rb49 line for our research.

Point 6: Section 2.5 subheading is confusing. Try to simplify it.

Response 6: We have simplified Section 2.5 subheading (Line 415-416) in revised manuscript.

Point 7: First paragraph of the Discussion (lines 373-387) (and other paragraphs as well) is very poorly written. There is a lot of spelling and grammar errors. Please, edit and improve it carefully.

Response 7: We appreciate your helpful suggestions. We have checked and improved the discusion in revised manuscript (Line 585-896).

Point 8: In Materials and Methods all paragraphs are presented twice. There are also some minor errors to be corrected.

Response 8: We have deleted the redundant Materials and Methods and corrected the errors.

Reviewer 3 Report

The paper contains some interesting data.  However in my opinion the paper needs to be improved before it could be considered for publication.

The inoculation process is not very well described.  It should be described in such a way that it is easy to repeat the experiments. For example density of inoculum etc would be helpful.

The description of  results of authophagosomes  presented in Suplementary fig 2 is very vague in the text. If those are similar to that observed for IR24 and IRBB3 plants it should be stated in the text.

Why some experiments were performed on both sets of plants and some on only one?

The description of RT-PCR conditions are also not sufficient > The is not enough information provided to allow for repetition of the experiment or even to judge if the experiments were performed correctly as in the cited paper different set of genes was analysed.

As 3-methyladenine can cause many otter effects that only inhibition of authophagy the conclusions that other changes  are authophagy related are a bit too far reaching.  Are you sure this is the results of authopahagy or just unrelated to aputhophagy inhibition of PI3K? The results would be the same  but you should be more cautious in your conclusions.  Maybe it would be more convinsintg if you shown all of the combinations on Fig 4

Did spraying plants with H2O cause any damage to plant not inoculated with bacteria?

At would suggest also language correction by native English speaker/ professional editing service  as some of the sentences are off for example line 365 “ Little H2O2 was accumulated”

Author Response

Response to Reviewer 3 Comments

Point 1: The paper contains some interesting data. However in my opinion the paper needs to be improved before it could be considered for publication.

Response 1: Thank the reviewer for the recognition of our work. We have revised the manuscript according to your suggestions.

Point 2: The inoculation process is not very well described. It should be described in such a way that it is easy to repeat the experiments. For example density of inoculum etc would be helpful.

Response 2: We greatly appreciate your helpful suggestions. We have added the detailed description of pathogen preparation and inoculation process in revised manuscript (Line 921-1155, Line 1148-1152, Line 1160-1165).

Point 3: The description of results of authophagosomes presented in Suplementary fig 2 is very vague in the text. If those are similar to that observed for IR24 and IRBB3 plants it should be stated in the text.

Response 3: We are very sorry for our mistakes of plant name notes in supplemental figure 2 legend. The plants belong to Rb49 and Mudanjiang 8 identical to the names in figure. We have corrected the plant names in description of supplemental figure in revised manuscript.

Point 4: Why some experiments were performed on both sets of plants and some on only one?

Response 4: There are two pairs of resistant and susceptible cultivars used in the manucript: indica cultivars IRBB3 and IR24, japonica cultivars Rb49 and Mudanjiang 8. Xa3/Xa26-mediated resistance to Xoo is influenced by genetic background and developmental stage (Cao et al. 2007). Japonica rice plants transformed Xa3/Xa26 with its native promoter show whole-growth-stage resistance (Sun et al.2004; Cao et al. 2007). But indica rice plants carrying Xa3/Xa26 only show adult stage (from tillering stage to grain filling stage) resistance (Cao et al. 2007). The expression of Xa3/Xa26 is relative higher throughout the growth stages in japonica plants, while is gradually increased from seedling stage to adult stage in indica plants, determines the molecular mechanism of Xa3/Xa26-mediated resistance modified by genetic background and development stage (Cao et al. 2007). We have added the description of the difference between those two pair cultivars in revised manuscript (Line 168-176). During tillering stage, the rice leaf becomes lumpy and brittle because of many big veins formation. Most solution runs away along the surface of rice leaf veins, and the leaves are easily broken off when the rice leaves at adult stage are inoculated by infiltration method. It is very difficult to infiltrate solution into rice leaf at adult stage. So, we have carried out most infiltrating inoculation on Rb49 plant at seedling stage that is resistant to Xoo and has flat and tough leaf. Furthermore, the leaf at adult stage is difficult to cut into ultrasections because of the thick cell wall of leaf cells. We have done most cellular expriments on Rb49 plant at seedling (four-leaf) stage. Rb49 plant is a stable transgenic line and widely used in Xa3/Xa36 function research (Sun et al. 2006; Cao et al. 2007, Deng et al. 2012; Li et al. 2014; Deng et al. 2018; Liu et al. 2018). Because the transgenic process could possibly affect gene expression patterns, we mainly used the non-transgenic cultivars IRBB3 and IR24 to analyze the expression patterns of ATG genes. At the same time, we have added some ATG genes expression analysis in Rb49 and Mudanjiang 8 plants in revised manuscript (supplemental figure 3). The expression patterns of some ATG genes in Rb49/Mudanjiang 8 plants were similar to that in IRBB3/IR24 plants (Line 280-282).

Point 5: The description of RT-PCR conditions are also not sufficient > The is not enough information provided to allow for repetition of the experiment or even to judge if the experiments were performed correctly as in the cited paper different set of genes was analysed.

Response 5: We greatly appreciate your helpful suggestions. We have added the detailed description of RT-PCR conditions in revised manuscript (Line 1184-1189).

Point 6: As 3-methyladenine can cause many otter effects that only inhibition of authophagy the conclusions that other changes are authophagy related are a bit too far reaching. Are you sure this is the results of authopahagy or just unrelated to aputhophagy inhibition of PI3K? The results would be the same but you should be more cautious in your conclusions. Maybe it would be more convinsintg if you shown all of the combinations on Fig 4

Response 6: We greatly appreciate your helpful comments. The class Ш phosphatidyinositol 3-kinase (PI3K) combines with ATG6 and other regulatory proteins to form a protein complex that is essential for the nucleation of autophagosome (Mizushima 2011; wang 2013). Although 3-methyladenine (3-MA) has other non-inhibiting-autophagy effects on mamalian cells, such as suppressing cell migration and invasion independently of its major ability to inhibit autophagy (Shingo et al. 2007). But, in plant cell, 3-MA inhibits autophagy by blocking the formation of autophagosomes and PI3K is essential for autophagy (Takatsuka et al. 2004). The autophagy inhibitor 3-methyladenine (3-MA) specifically inhibits PI3K activity and has been widely used to block the autophagy in plant cells (Inoue 2006; wang 2013; Fan 2019). We conclude that the cell death in Xa3/Xa26-mediated resistance belongs to autophagy-like cell death but not autophagy in our study. Thus, we hope our cautious conclusion could remove readers’ doubt about only 3-MA application on our study. We also the mechanism of 3-MA inhibiting autophagy (Line 256-259) in the Results.

Point 7: Did spraying plants with H2O cause any damage to plant not inoculated with bacteria?

Response 7: Spraying plants with H2O2 did not cause any damage to plants not inoculated with bacteria, which shows in column 1 of figure 5B as control. Spraying plants with H2O2 did not cause lesion on Rb49 and Mudanjiang 8 plants which inoculated with H2O. The empty column in figure 5B (note H2O+H2O2) shows this result that has been added to the revised manuscript (Line 422-424).

Point 8: At would suggest also language correction by native English speaker/ professional editing service as some of the sentences are off for example line 365 “Little H2O2 was accumulated”

Response 8: The native English speaker have helped us to check and revise all the language of manuscript. We have revised the sentence (Line 415-416) in revised manuscript.

Round 2

Reviewer 2 Report

The revised manuscript reads much better now. All of the raised questions have been addressed by the Authors and the manuscript changed accordingly.

Author Response

Point 1: The revised manuscript reads much better now. All of the raised questions have been addressed by the Authors and the manuscript changed accordingly.

Response 1: Thank the reviewer for the recognition of our work.

Reviewer 3 Report

The manuscript is much improved and I believe my previous concerns were resolved.

In RT- description please add the number of technical  replicates

I would also suggest some editing of the manuscript . The paper is much improved but  there are some awkward sentenced and I will definitely recommend additional language/ style  correction

Some examples:

e.g. Sentence starting from “but” in line 193

line 254 – “2.5. Little H2O2 accumulation in Xa3/Xa26-mediated resistance” - this sentence looks wrong to me. What did you want to say here ? that it was lower? Low? Somehow to me “little” is not explanatory enough

The same goes for Fig 5 especially first line. There I also think some word are missing form the end of this sentence. In vitro application maybe?

Line 310- sentence could be better written to eliminate double precipitate in the sentence- for example- “Calcium ion is reported to specifically react with potassium pyroantimonate to form as high electron-density calcium pyroantimonate precipitates under TEM”

Author Response

Response to Reviewer 3 Comments

Point 1: The manuscript is much improved and I believe my previous concerns were resolved.

In RT- description please add the number of technical replicates

Response 1: Thank the reviever for the recognition of our work. We have added the description of technical replicates in revised manuscript (Line 599-600).

Point 2: I would also suggest some editing of the manuscript. The paper is much improved but there are some awkward sentenced and I will definitely recommend additional language/ style correction

Response 2: We have re-checked the language and style in revised manuscript.

Point 3: e.g. Sentence starting from “but” in line 193

Response 3: We have edited the sentence in revised manuscript (Line 107-109).

Point 4: line 254 – “2.5. Little H2O2 accumulation in Xa3/Xa26-mediated resistance” - this sentence looks wrong to me. What did you want to say here? that it was lower? Low? Somehow to me “little” is not explanatory enough

Response 4: We are sorry for our confusing description in line 254. It should be low in this sentence. We have edited this sentence in revised manuscript (Line 273-274).

Point 5: The same goes for Fig 5 especially first line. There I also think some word are missing form the end of this sentence. In vitro application maybe?

Response 5: We have added the “in vitro H2O2 application “in this sentence of revised manuscript (Line 546-547).

Point 6: Line 310- sentence could be better written to eliminate double precipitate in the sentence- for example- “Calcium ion is reported to specifically react with potassium pyroantimonate to form as high electron-density calcium pyroantimonate precipitates under TEM”

Response 6: We have changed this sentence into “Calcium ion can specifically react with potassium pyroantimonate to form calcium pyroantimonate precipitates which show high electron-density under TEM [38]” in revised manuscript (Line 346-347).